# Dissecting Molecular Features of Gliomas: Genetic Loci and Validated Biomarkers

**DOI:** 10.3390/ijms21020685

**Published:** 2020-01-20

**Authors:** Antonietta Arcella, Fiona Limanaqi, Rosangela Ferese, Francesca Biagioni, Maria Antonietta Oliva, Marianna Storto, Mirco Fanelli, Stefano Gambardella, Francesco Fornai

**Affiliations:** 1IRCCS Neuromed, Via Atinense 18, 86077 Pozzilli, Italy; arcella@neuromed.it (A.A.); ferese.rosangela@gmail.com (R.F.); frbiagioni@libero.it (F.B.); olv78@homail.com (M.A.O.); marianna.storto@neuromed.it (M.S.); 2Department of Translational Research and New Technologies on Medicine and Surgery, University of Pisa, Via Roma 55, 56126 Pisa, Italy; f.limanaqi@studenti.unipi.it; 3Department of Biomolecular Sciences, University of Urbino “Carlo Bò”, 61029 Urbino, Italy; mirco.fanelli@uniurb.it

**Keywords:** glioblastoma, Next Generation Sequencing, Biomarkers, liquid biopsy, molecular diagnostics

## Abstract

Recently, several studies focused on the genetics of gliomas. This allowed identifying several germline loci that contribute to individual risk for tumor development, as well as various somatic mutations that are key for disease classification. Unfortunately, none of the germline loci clearly confers increased risk per se. Contrariwise, somatic mutations identified within the glioma tissue define tumor genotype, thus representing valid diagnostic and prognostic markers. Thus, genetic features can be used in glioma classification and guided therapy. Such copious genomic variabilities are screened routinely in glioma diagnosis. In detail, Sanger sequencing or pyrosequencing, fluorescence in-situ hybridization, and microsatellite analyses were added to immunohistochemistry as diagnostic markers. Recently, Next Generation Sequencing was set-up as an all-in-one diagnostic tool aimed at detecting both DNA copy number variations and mutations in gliomas. This approach is widely used also to detect circulating tumor DNA within cerebrospinal fluid from patients affected by primary brain tumors. Such an approach is providing an alternative cost-effective strategy to genotype all gliomas, which allows avoiding surgical tissue collection and repeated tumor biopsies. This review summarizes available molecular features that represent solid tools for the genetic diagnosis of gliomas at present or in the next future.

## 1. Introduction

The term “glioma” is commonly used to indicate solid tumors of the central nervous system (CNS) that may arise from and share histologic features with normal glial cells, namely astrocytes, oligodendrocytes, and ependymal cells. On this basis, gliomas are roughly classified as astrocytomas, oligodendrogliomas, and ependymomas, with each group including tumor subtypes that own different ranges of biological malignancy.

According to the traditional World Health Organization (WHO) classification, gliomas are roughly classified into four categories based on histological features (Figure 1). Grade I are solid and non-infiltrative tumors (pilocytic astrocytomas), while grades II–IV are diffuse infiltrating gliomas (DIGs) [1]. Grade II are commonly referred to as low-grade gliomas (LGGs), while the more aggressive tumors are referred to as high-grade gliomas (grade III and IV). Nonetheless, the WHO (IARC, Lyon) recommends avoiding the term “LGGs”, since it brings together a heterogeneous group of tumors that may significantly differ in biological properties, prognoses, and treatments [2,3].

Grade III gliomas are invasive and aggressive, owning quick progression leading to high lethality. Grade IV tumors, also known as “Glioblastoma”, are the most common and fatal adult brain tumors [4,5]. Glioblastoma is classified as either primary (de novo) or secondary in origin. Roughly 90% of glioblastomas develop rapidly de novo as primary glioblastomas, extremely aggressive tumors that affect mostly elederly patients [6]. Secondary glioblastomas (10%) are derived from previously lower-grade (WHO grades II or III) gliomas, namely diffuse or anaplastic astrocytomas, and they manifest mostly in younger patients [6] Secondary glioblastomas present at the onset with similar clinical and histological features as their primary counterparts, though different genetic/epigenetic profiles and molecular pathways are involved [6].

In the last few years, the application of Next Generation Sequencing (NGS) has revolutionized our understanding of somatic changes occurring within the cancer genome. This led to better understand the inter- and intra-individual heterogeneity of tumors, and to develop classification schemes aimed at characterizing the molecular subtype, considering dysregulated pathways that can be common or specific to a subset of cancer types [7,8,9,10,11,12,13,14,15].

To date, subtypes of glioma show different molecular and genetic profiles. According to the most recent classification (WHO 2016), histopathological analysis should always be accompanied by genetic screening [16]. This represents a substantial change towards integrating the concept of morphological and genetic classifications when translating specific cell tumor pathology into clinical practice [16]. For instance, the diagnosis of oligodendroglioma requires that tumor harbors both an *IDH1/2* mutation and a 1p/19q codeletion. Moreover, astrocytomas are featured by the *IDH1/2* mutation in the absence of 1p/19q codeletion, while often harboring inactivating mutations in α-thalassemia mental retardation X-linked *(ATRX)* and tumor protein *p53* (*TP53)* genes.

Methylation profiling may be added to histological and standard genetic approaches to classify brain tumors, potentially refining future classifications [6]. In this scenario, tumor classification according to molecular subtypes represents a diagnostic, prognostic, and potentially therapeutic marker [3,7,8,9,10,17,18,19,20,21]. As a consequence, these molecular markers may overwrite the histological phenotype, which may significantly impact treatment options in each patient.

This review summarizes those main molecular and genetic features of gliomas that may represent solid tools for the genetic diagnosis at present and in the next future.

## 2. Germline Features and Loci Influencing the Risk of Glioma

The risk of gliomas is consistently elevated in first-degree relatives of patients with gliomas and other primary brain tumours. Therefore, a great effort has been made to understand the genetics of gliomas [22].

Most cases cannot be explained by causes related with endogenous or exogenous factors. In fact, the only generally accepted and well-defined risk factors are high doses of ionizing radiation and rare genetic syndromes. Unfortunately, they can only explain a small percentage of all gliomas.

Except for a few rare mendelian cancer predisposition syndromes (i.e., Li Fraumeni syndrome, Neurofibromatosis), the genetic basis of inherited susceptibility to gliomas is currently undefined given the unlikeness of a disease susceptibility model that is solely based on high-risk mutations. In fact, as demonstrated in other cancer diseases, much of the inherited risk is likely to be the result of the co-inheritance of common multiple low-risk variants. To this aim, genome-wide association studies (GWAS) and additional fine-mapping identified some common germline genetic variants associated with an increased risk of glioma [23,24,25,26,27,28,29,30,31,32,33,34]. To date, more than 25 genetic loci have been associated with an increased risk of developing glioma in adulthood [23,24,25,26,27,28,29,30,31,32]. Most genes located within these loci are affected by somatic mutations occurring in gliomas, namely cyclin-dependent kinase inhibitor 2A and B (*CDKN2A*, *CDKN2B*), epidermal growth factor receptor (*EGFR)*, telomerase reverse transcriptase (*TERT), TP53,* pleckstrin homology-like domain family B member 1 (*PHLDB1),* and regulator of telomere elongation helicase 1 (*RTEL1)* [26,27,28,35,36,37,38]. The first germline studies identified a locus on chromosome 9p21, encompassing the *CDKN2A* (MIM number 600160) and *CDKN2B* (MIM number 600431) tumor suppressor genes, which have an established role in glioma development. In keeping with this, homozygous deletion in *CDKN2A* is detectable in approximately 50% of tumors [7], and the loss of *CDKN2A* expression is linked to poor prognosis. Furthermore, *CDKN2A* germline mutations are responsible for the melanoma-astrocytoma syndrome (MIM number 155755), and genetic variants close to both *CDKN2A* and *CDKN2B* genes (on the chromosomal locus 9p21) are known to increase the risk for glioma, basal cell carcinoma, and melanoma [35].

Correlations between germline and somatic variants suggest that an association between germline genetic variation and environmentally-induced molecular alterations could diverge as a key to define a single molecular event in different gliomas. This is consistent with germline variants at 8q24.21, which are associated with *IDH1- IDH2* mutated astrocytoma and oligodendroglial tumors [3].

Some germline genetic variants are associated with tumor grade. For example, high-grade gliomas are associated with risk variants in *RTEL1, CDKN2B,* and *TERT* [32,38], while low-grade gliomas with *IDH* mutation-1p/19q codeletion are associated with risk variants in *CCDC26* and *PHLDB1* regions [17,32,38]. Although these germline loci confer increased individual risk, none of them does represent, per se, a reliable association to be used in clinical routine.

## 3. Somatic Molecular Features for Glioma Classification

### 3.1. Molecular Features of Astrocytoma and Oligodendroglioma

Diffuse gliomas (DGs) of the astrocytic and oligodendroglial lineages (grade II and III) are characterized by frequent *IDH* mutations (Figure 2A). *IDH1* encodes for the isocitrate-dehydrogenase enzyme 1, which catalyzes oxidative carboxylation of isocitrate to α-ketoglutarate, thus, generating nicotinamide adenine dinucleotide phosphate hydrogen (NADPH) [39]. Mutations in *IDH1* or its homolog 2 (*IDH2*) have been identified as early molecular events in the development of astrocytomas and oligodendrogliomas, and they occur in various types of malignancies, including IDH-mutant glioblastoma (grade IV) (Figure 2A) [7,40,41]. These mutations affect the R132 codon of *IDH1* or the corresponding R172 codon in its homolog *IDH2* (p.R172K, p.R172W, and p.R172M) [39,41,42,43,44,45,46,47,48,49,50], which fall in catalytically-active sites of these enzymes [42,43].

Although both types of gliomas contain *IDH1/2* mutations, in astrocytic gliomas, *IDH* mutations are typically associated with mutations in *TP53* and *ATRX* genes [7,50] (Figure 2A). In fact, the absence of the ATRX protein and the abundance of p53 protein are required for the diagnosis of astrocytoma.

Traditional methods testing p53 status consist of p53 accumulation in the cell nuclei detected via immune-staining. Later on, sequence analysis was focused on hotspot mutations located in exons 5–8, although recent studies suggest that *TP53* carries mutations also outside its mutational hotspots.

Loss-of-function mutations in *ATRX* (SWI/SNF chromatin re-modeler gene) co-occur with mutations in *TP53* and *IDH1*/*IDH2* genes in adults or H3.3 histone monomers genes (*H3F3A* and *HIST1H3B*) in children [51,52,53,54]. Recent papers suggest that *ATRX* deficiency produces deleterious effects on genomic integrity through an alternative lengthening of telomeres (ALT) that impairs the physiological arrest of cell division [55,56].

Similarly to astrocytomas, oligodendrogliomas have *IDH* mutations associated with co-deletion of chromosomal arms 1p and 19q (Figure 2A) [54,57]. According to the 2007 WHO classification, oligodendrogliomas could be diagnosed only when typical histological features are associated with the recently defined “oligodendroglioma, *IDH*-mutant and 1p/19q codeleted”, a molecular condition that requires the concomitancy of both *IDH1* or *IDH2* mutations and 1p/19q co-deletion. Oligodendrogliomas often feature mutations in the Drosophila homolog of capicua transcriptional repressor (*CIC*) gene and the *TERT* promoter [54,57,58,59], which encodes for the catalytic subunit of telomerase [58]. The two most common mutations in the *TERTp* are p.C228T and p.C250T, which are located upstream of the *TERT* ATG start site. These mutations produce *TERT* activation, which is responsible for the growth properties of the tumor cells, indicating the importance of its role in cancer development [60,61] (Figure 2A).

According to the WHO 2016 classification, assessment of p*TERT* mutation may assist histological diagnosis. In fact, in LGGs the presence of a p*TERT* mutation in association with *IDH* mutation and 1p/19q co-deletion may represent a diagnostic surrogate for oligodendroglia lineage. Similarly, the presence of a p*TERT* mutation without *IDH* mutation and 1p/19q co-deletion suggests an aggressive clinical course [62]. It is known that the effect of *pTERT* mutations is inversely related to mutations found in the *IDH1* gene [34], and this is strongly related to a favorable prognosis [28].

### 3.2. Molecular Features of Primary Glioblastomas

The genetic heterogeneity of glioblastoma prompted a molecular characterization to identify various molecular subtypes. Primary glioblastomas typically lack *IDH* mutations [17], while possessing severe dysregulations of specific signaling pathways (Figure 2B). This is best exemplified by alterations in tyrosine kinase receptor (RTK)/RAS/PI(3)K signaling leading to *EGFR* amplification. *EGFR* is a proto-oncogene, which encodes for a transmembrane receptor of the ERBB family [63]. A quite common activating *EGFR* mutation consists of exons 2–7 deletion leading to a truncated *EGFR* variant III (EGFRvIII), which is often described in glioblastoma [64,65]. This deletion translates into a truncated extracellular domain with ligand-independent constitutive activity due to the loss of amino acids 6–273 producing a junction site with a new glycine residue (amino acids 5–274) [66]. Among *EGFR* mutants, *EGFRvIII* represents a late event following wild type *EGFR* amplification [67]. While the majority of glioblastoma cells express either *EGFR* or *EGFRvIII*, a small fraction of tumor cells co-express *EGFR* and *EGFRvIII* (Figure 2B). In addition, 24% of glioblastoma samples have point mutations in the extracellular region of *EGFR* which keep EGFR in active conformation [64,65].

Primary glioblastomas also carry mutations in the phosphatase and tensin homolog (*PTEN*) gene and in the *TERT* promoter. In addition, monosomy of chromosome 10, gain of chromosome 7 or 7q, and loss of 9p involving the *CDKN2A/B* loci often occur [68].

### 3.3. Glioma Epigenetics

Over the last decade, mutations in epigenetic regulator genes have been identified as key drivers for specific subtypes of glioma with distinct clinical features [69]. This is the case of mutations in *IDH1* or *IDH2* in lower-grade gliomas, and histone 3 *(H3F3A* and *HIST1H3B*) mutations in pediatric high-grade gliomas that are also associated with specific patterns of DNA methylation [70].

The *O*^6^-alkylguanine DNA alkyltransferase gene (*MGMT)*, a DNA repair gene, is the most prominent epigenetic biomarker in gliomas (Figure 2B) and its alterations play a central role in GMB classification, treatment, and survival outcomes.

In line with this, *MGMT* methylation status has become the first predictive biomarker in neuro-oncology [20,71,72,73,74,75,76]. In fact, promoter methylation-dependent silencing of *MGMT* predicts sensitivity to alkylating agent therapy [73,74]. *MGMT* silencing may blunt therapeutic efficacy by hampering the repairs of O^6^-methylguanine, a toxic damage induced by alkylating agents.

Therefore, GMB patients with unmethylated *MGMT* do not profit much from the addition of temozolomide either as a concomitant drug or as an adjuvant to radiotherapy [20]. Routine determination of the *MGMT* status allows stratified treatment, thus, it should be used to select glioblastoma patients for clinical trials allowing to omit temozolomide in their treatment arm [75,76].

Correlations between *IDH1* mutation, *MGMT* promoter methylation and survival outcomes have been analyzed [77]. The role of the *MGMT* methylation status on benefit from temozolomide in *IDH* mutant (*IDHmt*) lower-grade gliomas is less clear [78], although most cases (>80%) show methylation at the *MGMT* promoter. However, unlike glioblastomas lacking one copy of chromosome 10 where *MGMT* is located, *IDHmt* lower-grade gliomas usually retain both copies and *MGMT* may not be completely silenced. This results in residual repair capacity of *MGMT* conferring slight resistance to temozolomide therapy [79]. In routine tests, methylation-specific PCR is genererally employed to determine the *MGMT* methylation status [80]. An *MGMT* status classifier (MGMT-STP27) is available for samples analyzed at Illumina DNA methylation platform (HM27 K, HM450 K, EPIC) [81], which is routinely applied in clinical trials [82].

## 4. Somatic Molecular Features and Copy Number Variation

Recurrent Copy Number Variations (CNVs) have been reported in gliomas, where various chromosome regions are involved following several combinations, suggesting a strong genomic instability. The induction of a genomic instability phenotype is a crucial and early event in carcinogenesis [83,84].

Three types of genomic instability, namely microsatellite (MIN), chromosomal (CIN), and point mutation instability (PIN), may be manifestations of a mutator phenotype [85]. CIN is defined as an increased propensity to acquire chromosome aberrations following dysfunctional processes in chromosome replication, repair, or segregation [86].

CIN, which represents a persistently high rate of chromosome mis-segregation, is responsible for CNV of whole or part of chromosomes, thus, representing a major source of genetic heterogeneity. In fact, CIN constantly causes modifications of karyotypes that result in extensive heterogeneity within cells [87,88].

The 1p/19q co-deletion likely represents the best characterized and most studied marker [89]. This codeletion consists of the complete deletion including both the short arm of chromosome 1 (1p) and the long arm of chromosome 19 (19q). This deletion is produced by a balanced whole-arm translocation of chromosomes 1/19, which leads to the formation of two derivative chromosomes, where the chromosomes composed of 1p and 19q (der [1,19] [p10; q10]) is typically lost [90,91].

The 1p/19q co-deletion represents the molecular marker of oligodendrogliomas, a subtype of primary brain tumors accounting for approximately 15% of all DGs in adults [92,93]. According to the 2016 WHO classification, detecting 1p/19q co-deletion is essential for the neuropathology of gliomas owning an oligodendroglioma-like morphology [94]. Virtually, 1p/19q co-deleted tumors harbor an accompanying *IDH1/IDH2* mutation [95,96]. Other common molecular alterations co-occurring with 1p/19q co-deletion include variants in *TERT*, *CIC*, (far upstream element-binding protein 1) *FUBP1* as well as *MGMT* promoter methylation [56,97].

## 5. Diagnostic Approaches and NGS Application

In the past, molecular assays for glioma required the application of different techniques such as immunohistochemistry (IH) with mutation-specific antibodies, conventional Sanger sequencing or pyrosequencing, fluorescence in situ hybridization (FISH), and microsatellite analyses. IH has been widely used for the detection of EGFRvIII in tumors with *EGFR* rearrangements, as well as for ATRX, TP53, and IDHR132.

Array comparative genomic hybridization (aCGH), a technique enabling high-resolution, genome-wide screening of CNVs, has been fundamental to detect complex genomic alterations that involve gliomagenesis, including sequence changes, CNVs, and epigenetic modifications [98].

From what is reported, it appears that multiple testing is required to support a complete diagnostic workup of glioma, where biomarkers can be individually assessed by different techniques. Unfortunately, this implies that analyses of each biomarker need to be carried out sequentially, making the process time-consuming [99].

Therefore, diagnostics of gliomas, which, according to the WHO 2016 classification, also includes the assay of CNVs, has prompted novel strategies enabling the concomitant analysis of several biomarkers. In the last years, NGS-based approaches were progressively used in the diagnosis of gliomas considering those genes that are commonly altered. When applying NGS assay for the routine diagnosis of CNS tumours, several issues need to be considered: (i) The assay should identify single nucleotide variations, insertions and deletions, complex genetic alterations (gene fusions, CNVs) in a set of genes, and mutational hotspots. (ii) The analysis should be reliable in formalin-fixed paraffin-embedded (FFPE) tissues. (iii) The assay should require a minimal amount of DNA/RNA since resection is not feasible in many patients [17].

To this aim, several targeted NGS panels were implemented for selected genetic regions, focused on point mutations, insertions and deletions, CNV changes, and gene fusions [100,101,102].

Nikiforova et al. set up a single workflow analyzing 30 genes for SNVs and indels, 24 genes for CNVs, and 14 types of structural alterations in *BRAF*, *EGFR*, and *FGFR3* genes [17]. Ballester et al. used a more extensive NGS panel of 50 genes, and found that the most clinically relevant genes were *IDH1, IDH2, TP53, PIK3CA, BRAF, EGFR, PDGFRA,* and *FGFR1/2/3* [103].

Zacher et al. used a 20-gene panel for an integrated histological and molecular diagnosis of 111 diffuse gliomas, allowing a re-classification of oligoastrocytoma and glioblastoma by *IDH*-status and identification of tumours with *H3F3A* mutations [104].

Although these approaches based on selected genes represent a promising diagnostic tool in clinical routine, they show a limitation in detecting rare genetic aberrations that are known to contribute to the pathogenesis of gliomas. This is the case of tuberous sclerosis 1/hamartin (*TSC1*) or tuberous sclerosis 2/tuberin (*TSC2*) mutations occurring in subependymal giant cell astrocytomas [105], as well as H3 histone family member A/B (*HIST1H3A/B*) and activin receptor type 1 (*ACVR1*) mutations in diffuse intrinsic pontine gliomas [106,107].

To improve the identification of rare variants, Virk et al. analyzed the coding sequences of 409 genes implicated in cancer. Unfortunately, variants were identified in 13 (3%) out of the 409 genes, suggesting a futility in assaying too many genes [108].

These approaches, considering targeted or extended genes panel, reliably assess key mutations such as CNVs, including 1p/19q codeletion and *EGFR* gene amplification, *CDKN2A* homozygous deletion, and *PTEN* loss [109]. Despite a number of advantages, NGS carries inherent limits for specific biomarkers such as loss of *ATRX* expression; in this respect NGS may have sometimes less sensitivity than IH.

## 6. Liquid Biopsies in Patients With Diffuse Glioma

The term Circulating Nucleic Acids refers to segments of DNA/RNA found in the bloodstream or other body fluids. These molecules were identified as all cell-free entities, attached to lipid or protein structures, and as the content of circulating extracellular vesicles (EVs) and blood platelets [110,111]. Cancer cells release circulating tumor DNA (ctDNA) through yet unclear mechanisms, although apoptosis or necroptosis are suggested to be the principal source [112,113,114]. The analysis of ctDNA and other nucleic acids in body fluids is also known as “Liquid Biopsy”. This approach considers the blood as the main source of ctDNA since serum shows higher background levels probably due to contamination with DNA that is released from immune cells lysing during the clotting process. Therefore, plasma is generally preferable to serum [114].

In support of this, ctDNA has been detected, analyzed, and quantified in plasma of patients with different types of cancer [114,115,116]. Tagged-amplicon deep sequencing (TAm-Seq) allowed detecting blood-ctDNA mutations in about 2% of gliomas [115]. The most common variants detected in the blood-ctDNA of glioma patients are *IDH1, EGFR, p53,* and *PTEN* [114]. Nonetheless, the identification of ctDNA in the blood of patients with primary brain tumours is challenging due to the presence of the blood–brain barrier (BBB), which may impede the free circulation of DNA [116,117,118]. Therefore, the use of alternative fluids such as the cerebrospinal fluid (CSF) may provide an alternative way to genotype gliomas [117].

To this aim, Wang et al. analyzed 35 primary CNS malignancies using targeted or genome-wide sequencing. They detected CSF-ctDNA in 74% of cases, and mutations were detected in each patient [118]. De Mattos-Arruda et al. found mutations in *IDH1/2, EGFR, PTEN, FGFR2,* and *ERBB2* genes, and they showed that CSF-ctDNA undergoes dynamic changes that are correlated with the treatment course of patients with glioblastoma [117]. Recently, the presence of ctDNA was detected in about 50% of 85 patients with gliomas, and it was associated with disease burden and adverse outcome [119]. This study demonstrated that the genomic features of ctDNA in CSF from patients with gliomas closely resembled the genomes of tumour biopsies. In this scenario, liquid biopsy allows monitoring the evolution of the cancer genome through a minimally invasive technique, which may provide, in the future, the opportunity of genotype-directed therapies for gliomas.

Of relevance, ctDNA could be used to identify the methylation status of gene promoters involved in CNS tumors. Indeed, in both CSF and tissue from glioblastoma patients, Chen et al. identified promoter hypomethylation of *MGMT, p16INK4a, TIMP-3,* and *THBS1* genes [120].

Liquid biopsy has allowed identifying additional components in the CSF of glioblastoma patients. This is the case of CSF-isolated EVs which allowed the detection of *IDH1* mutations in five out of eight patients using beads, emulsion, amplification, magnetics (BEAMing) and Droplet Digital PCR (ddPCR) techniques. These outcomes match with the corresponding *IDH1* mutations detected in tumor tissues [121]. In addition, circulating as well as EV-derived miRNAs have been detected in CSF samples [122,123], suggesting that they might serve as CSF biomarkers to diagnose and monitor therapy response in glioblastoma patients [124]. In support of this, the levels of a panel of nine miRNAs detected in the CSF of glioblastoma patients, including miR-21, miR-218, miR-193b, miR-331, miR374a, miR548c, miR520f, miR27b, and miR-30b, were found to correlate with those detected from tissue biopsies [125].

## 7. Merging Genomics and Immunology: A Glance at Future Perspectives

Investigating the expression patterns and clinical characteristics of immune checkpoint proteins as well as the specific genomic alterations that produce immunogenic neo-antigens/peptides may be helpful for expanding our understanding of antitumor immunotherapy in gliomas [126,127]. Indeed, this is the era of personalized medicine where the marriage between genomics and immunology dictates the successes of immunotherapy in cancer, from checkpoint inhibitors to CAR T cells [126,127].

Recently, 41 immune prognostic genes were identified in glioma patients based on the *IDH1* mutation status [128]. High-risk LGG patients express high levels of (CD28/cytotoxic T-lymphocyte-associated protein 4) CTLA-4 and programmed death protein-1 (PD-1), and they possess high levels of infiltrating B cells, CD4+ T cells, CD8+ T cells, neutrophils, macrophages, and dendritic cells [128]. CTLA-4 is an immune checkpoint protein that negatively regulates T cell-mediated immune responses. Higher expression levels of CTLA-4 occur in patients with higher grade *IDH-wt* gliomas compared with patients bearing lower grade, *IDH-mt* gliomas [126]. A positive correlation exists between CTLA-4 expression and specific marker genes of immune cells, including CD8^+^ T cells, regulatory T cells, and macrophages. Thus, the higher the CTLA-4 expression in the glioma microenvironment, the greater the immune cell infiltration, and the worse the prognosis. In fact, patients with glioma possessing lower CTLA-4 expression levels exhibit significantly longer overall survival [126].

In glioma patients, a correlation has also been identified among *EGFR* mutation, shorter survival time and increased infiltration of immune cells along with PD-L1 expression [129]. Programmed death-ligand 1 (PD-L1) is the ligand of PD-1, a cell surface receptor belonging to the extended CTLA-4 family of T cell regulators. As recently reviewed, the expression levels of PD-L1 are positively correlated with glioma grades and negatively correlated with the prognosis of glioma patients [130]. PD-L1 levels are also associated with glioma genotypes, with PD-L1 expression being significantly lower in *IDH-mt* compared with *IDH-wt* gliomas [130]. In detail, PD-L1 expression is negatively correlated with PD-L1 promoter methylation level, which is consistently higher in *IDH-mt* compared to *IDH-wt* gliomas [130].

Thus, classifying glioma patients into subgroups based on distinct immunophenotypes, genotypes, and outcomes may be a clinically promising biomarker, which may improve prognosis and foster personalized therapies for gliomas.

## 8. Conclusions

Genetic analysis is key to identify molecular features of glioma tissue, to identify germline loci that contribute to individual risk for tumor development, and to select potential biomarkers useful for clinical applications.

Except for a few rare mendelian cancer predisposition syndromes, the genetic basis of inherited susceptibility to gliomas is currently undefined since none of the germline loci clearly confer an increased risk of developing the disease. On the contrary, somatic mutations identified within the tissue define tumor genotype as well as glioma subtypes, focusing on dysregulated pathways that can be either common or specific to a subset of cancers.

These molecular markers may overwrite the histologic phenotype, thus, representing a valid diagnostic and prognostic markers for glioma classification and guided therapy. To date, most data have been produced by using several molecular approaches like Sanger sequencing or pyrosequencing, fluorescence in-situ hybridization, microsatellite analyses and immunohistochemistry. In the last years, the implementation of NGS has revolutionized our understanding of cancer somatic genome changes, as well as the inter- and intra-individual heterogeneity of tumors. This technique is now under evaluation as an all-in-one diagnostic tool aimed at detecting both DNA copy number variations and mutations in glioma. NGS approaches based on selected genes represent a promising diagnostic tool in clinical routine; yet, they show a limitation in detecting rare genetic aberrations that are known to contribute to the pathogenesis of gliomas. Unfortunately, protocols based on the evaluation of hundreds of genes witness for futility in assaying too many genes. Despite a number of advantages, NGS carries inherent limits for specific biomarkers, showing less sensitivity than IH.

NGS approach is also widely used in Liquid Biopsy to detect ctDNA within the CSF from patients affected by primary brain tumors. Such an approach is providing an alternative cost-effective approach to genotype all gliomas, while avoiding surgical tissue collection and repeated tumor biopsies. CSF-ctDNA undergoes dynamic changes in CSF which are correlated with the treatment outcomes of patients with glioma, and genomic features of glioma CSF-ctDNA closely resemble the genomes of tumour biopsies. In this scenario, liquid biopsy allows monitoring the evolution of the glioma genome through a minimally invasive technique, giving the future opportunity of genotype-directed, tailored therapy for glioma patients. In this context, the marriage between genomics and the immune system may provide clinically revelant biomarkers improving glioma patients’ stratification and treatment.

## Figures and Tables

**Figure 1 ijms-21-00685-f001:**
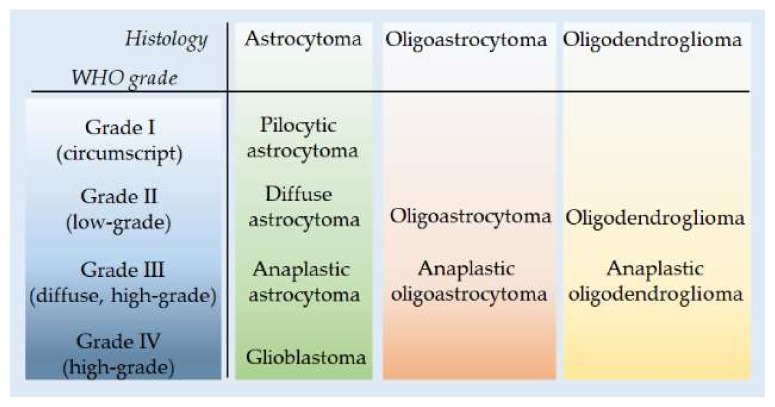
Histological classification of gliomas according to the World Health Organization WHO (I–IV) grades.

**Figure 2 ijms-21-00685-f002:**
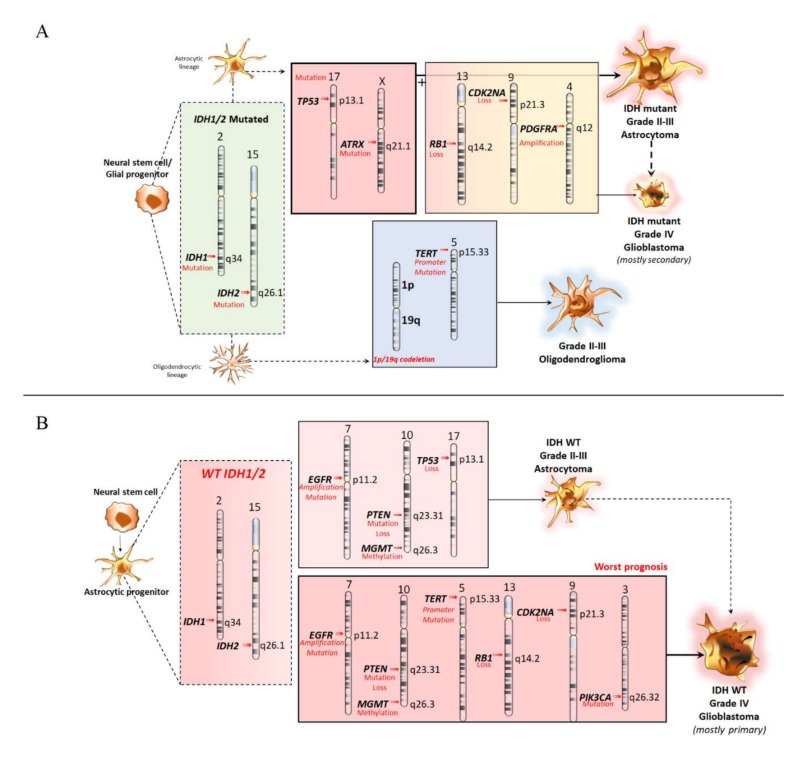
Genetic Biomarkers for *IDH1/2*-mutated (**A**) and *IDH1/2*-wild type (**B**) Astrocytomas and Oligodendrogliomas This cartoon roughly summarizes the main somatic gene mutations that are key in the classification of diffuse gliomas (DGs). (**A**). Biomarkers for astrocytomas/oligodendrogliomas featuring *IDH1/2* mutations. Grade II-III astrocytomas are classified based on the occurrence of mutations within *IDH1/2* along with *TP53* (17p13.1) and *ATRX* (α-thalassemia mental retardation X*-linked,* Xq21.1). Grade IV astrocytoma (glioblastoma) arise mostly secondarily to lower-grade astrocytoma and, to a lesser extent, primarily from additional mutations occurring within *CDKN2A*/*B, RB1,* and *PDGFRA* (platelet derived growth factor receptor alpha 4q12). The diagnosis of grade II-III oligodendroglioma requires the presence of a *1p/19q* codeletion and/or *TERT* promoter mutation besides *IDH1/2* mutation. (**B**). Biomarkers for astrocytomas in the absence of *IDH1/2* mutations. In the absence of mutations within *IDH1/IDH2* genes (2q34 and 15q26.1, respectively), the classification of grade II–III and/or grade IV astrocytoma (mostly primary glioblastoma) is based on frequently occurring somatic mutations within the following genes: *PIK3CA* (phosphatidylinositol-4,5-bisphosphate 3-kinase catalytic subunit alpha, 3q26.32); *TERT* (telomerase reverse transcriptase, 5p15.33); *EGFR* (epidermal growth factor receptor, 7p11.2); *RB1* (retinoblastoma ranscriptional orepressor 1, 13q14.2); *PTEN* (phosphatase and tensin homolog 10q23.31); *CDKN2A*/*B* (cyclin-dependent kinase inhibitor 2A/B, 9p21.3); *TP53* (tumor protein *p53,* 17 p13.1); *MGMT* (O6-alkylguanine DNA alkyltransferase, 10q26.3) methylation status may serve as a predictive biomarker of chemotherapeutic susceptibility.

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
