# Peer review of "Dissecting Molecular Features of Gliomas: Genetic Loci and Validated Biomarkers"

_ijms, 2020, doi:10.3390/ijms21020685_

Round 1

Reviewer 1 Report

In their manuscript „Dissecting genetic biomarkers in gliomas” Arcella and co-worker give a brief overview not as much of biomarkers, but mainly about genetic and epigenetic markers of disease before focusing in the final section of their paper on liquid biopsies. As a whole their paper is an enjoyable and well-informed read, but contains a few passages which are confusing and could be further improved.

Major points:

As already mentioned, my understanding of a biomarker is that it either contributes to disease stratification or has a predictive value regarding therapy response. Disease defining alterations are not per se biomarkers, yet a lot of themes discussed in this paper are exactly that. Furthermore, epigenetic markers are also addressed, which contradicts the title. Therefore, my suggestion would be: change the title. I find several passages unclear, starting with the first few sentences of the abstract, I am not sure what the authors are aiming for, I am assuming page 1, line 15: which rules *them* out *as* genetic biomarkers? Other aspects which should be clarified: Page 1, line 40: Explain the different properties of the gliomas if you use them as examples for different properties Figure 1 (replace cartoon with scheme?), part A is not clear what the difference between Grade IV and II and III is, generally not easy to understand, part B is easier. What is the take home of the paragraph on page 7, line 264-271? I cannot follow the conclusion of paragraph on page 7, line 278-282. Is 3% not cost/benefit relevant enough for 409 genes? How many samples were analysed, etc (also, add comma after Therefore in 278). Section 6, in many respects the heart piece of the paper, is unclear whether it is dealing solely with GBM or all DGs, the focus seems to shift. Section 6 ends with a comparison between CSF and blood, suggesting that DNA from blood is a bad surrogate for GBM. This needs to be emphasized and leads to my point 3: Please add a conclusion, what are the consequences and suggestions which the authors draw from their work. Many important issues, not least CSF vs blood, are raised, but the paper just fizzles out, thus the paper has no overriding arche. I find it difficult to follow how many and which gliomas are discussed, a table summarizing all (major) forms of glioma with defining features would be helpful. Now, I am not sure about this point, but while reading the paper I wondered whether a diagnostic flowchart would supplement the text, be showing how one derives at a conclusion: IDH1 negative/positive? Depending on yes, or no, leading to TP53 mutated Yes/No? etc. This might distract from the overall focus of the paper, it will also further distract from histological definitions, which should at least be mentioned in the paper regardless whether this suggestion is implemented or not

Minor points:

Page 1, line 42: owing?

Page 1, line 44: Glioblastoma multiforme is more commonly referred to as glioblastoma

Page 2, line 46: Mention that secondary GBMs are surprisingly more common than primary.

Page 2, line 70/71: Language needs to be cleaned up.

Page 2, line 73: and not sand

Page 2, line 89: 600431 link dead.

Page 5, line 149: Similarly is better than Likewise

Page 5, line 166: IDH1 is not completely in italics, reference 32 contains half a link

Page 5, line 176: Make clear that the vIII mutation is activating!

Page 5, line 182: The fullstop is a link?

Page 5, line 196: o6-methylguanine is not the most toxic damage, it is just damage that is not repaired by BER (which is active in GBM). Also, TMZ currently rather hotly debated topic.

Page 6, line 200: MGMT status is not often used (it should be!), two references one of which not really on topic do not support that statement.

Page 6, line 206: Replace placed with located

Page 7, line 247: From what *is* reported

Page 7, line 286: may own sometimes?

Page 8, line 308: add *ctDNA found in blood of* between in and patients

Page 9, line 353: diventa?

Several statements need a reference added, several abbreviations need to be defined at first use. Abbreviation list at the end should be alphabetical.

Author Response

We would like to thank the referee for the important suggestions that allowed to improve the quality of the paper. We edited pictures, we added the conclusions, and we modified the title and paragraphs concerning the term of biomarkers, mainly substituted with "molecular features".

Here we have the opportunity of uploading just one file. We are uploading the doc file with active revision. We are sending the whole file including the new pictures  to the editorial office

Therefore, my suggestion would be: change the title.

       the title has been changed

page 1, line 15: which rules *them* out *as* genetic biomarkers?

       the sentence has been edited

Page 1, line 40: Explain the different properties of the gliomas if you use them as examples for different properties

       the introduction has been deeply edited, and a figure has been added

Figure 1 (replace cartoon with scheme?), part A is not clear what the difference between Grade IV and II and III is, generally not easy to understand, part B is easier

        the figure has been changed and improved with a second figure

What is the take home of the paragraph on page 7, line 264-271? I cannot follow the conclusion of paragraph on page 7, line 278-282. Is 3% not cost/benefit relevant enough for 409 genes? How many samples were analysed, etc (also, add comma after Therefore in 278).

       the section has been edited following the reviewer's suggestion

Section 6, in many respects the heart piece of the paper, is unclear whether it is dealing solely with GBM or all DGs, the focus seems to shift. Section 6 ends with a comparison between CSF and blood, suggesting that DNA from blood is a bad surrogate for GBM. This needs to be emphasized and leads to my point 3:

        Section 6 was completely rewritten following the reviewer's suggestions

Please add a conclusion, what are the consequences and suggestions which the authors draw from their work.

       Conclusions have been added to the paper

Many important issues, not least CSF vs blood, are raised, but the paper just fizzles out, thus the paper has no overriding arche.

       section 6 was completely rewritten following the reviewer's suggestions

a table summarizing all (major) forms of glioma with defining features would be helpful.

        Figure 1 has been added to explain all forms of glioma based on WHO                guidelines

Minor points:

Page 1, line 42: owing? it was edited

Page 1, line 44: Glioblastoma multiforme is more commonly referred to as glioblastoma it is ok

Page 2, line 46: Mention that secondary GBMs are surprisingly more common than primary. it was edited

Page 2, line 70/71: Language needs to be cleaned up. it was edited

Page 2, line 73: and not sand it is ok

Page 2, line 89: 600431 link dead. it was added

Page 5, line 149: Similarly is better than Likewise it was edited

Page 5, line 166: IDH1 is not completely in italics, reference 32 contains half a link it was edited

 Page 5, line 176: Make clear that the vIII mutation is activating! it was edited

Page 5, line 182: The fullstop is a link? it was edited

Page 5, line 196: o6-methylguanine is not the most toxic damage, it is just damage that is not repaired by BER (which is active in GBM). Also, TMZ currently rather hotly debated topic. it was edited

Page 6, line 200: MGMT status is not often used (it should be!), two references one of which not really on topic do not support that statement. The referenced was changed

Page 6, line 206: Replace placed with located it is ok

Page 7, line 247: From what *is* reported it is ok

Page 7, line 286: may own sometimes? it is ok

Page 8, line 308: add *ctDNA found in blood of* between in and patients it is ok

Page 9, line 353: diventa? it is ok

Several statements need a reference added, several abbreviations need to be defined at first use. Abbreviation list at the end should be alphabetical.

The referenced was added

Reviewer 2 Report

The review by Arcella and colleagues, focused on describing all the genetic biomarkers in gliomas, is complete and well written. The authors have also addressed the liquid biopsy, currently one of the most relevant tool used to improve the molecular characterization of gliomas, and to discover novel therapeutic targets for potential personalized treatments.

The authors should consider an additional paragraph addressing the immune-biomarkers in gliomas, in relationship with genetic and diagnostic biomarkers, and their therapeutic perspectives.

More emphasis on the translation relevance of predictive and prognostic biomarkers should be considered.

Author Response

We would like to thank the referee for the important suggestions that allowed to improve the quality of the paper. We tried to better describe the translation relevance of predictive and prognostic biomarkers.

Regarding the second question about an additional paragraph addressing the immune-biomarkers in gliomas, we tried to study all the papers regarding this topic. Unfortunately, after several efforts, we realized that we don't have the proper background to dissect this topic.

Therefore, we kindly ask the referee if he can suggest us another way to further improve the quality of the paper, and sure we will try to follow the recommendations

Round 2

Reviewer 1 Report

The authors have improved on their paper and I would only suggest a few minor tweaks.

English could be polished a bit more.

When arguing with WHO classifications Glioblastoma should really be used and not Glioblastoma multiforme.

Based on typo by me (i.e. mea culpa): Primary glioblastoma are more common than secondray, by far! (90% to 10%, add refernce)

Page 6/7 there is a paragraph in bold which shouldn't be.

Author Response

Reviewer 1

The authors have improved on their paper and I would only suggest a few minor tweaks.

Q1. English could be polished a bit more.

Reply: We wish to thank the reviewer for the suggestions to improve our manuscript. Accordingly, with the help of a native English speaker, the manuscript was carefully checked and revised concerning the English language throughout.

Q2. When arguing with WHO classifications Glioblastoma should really be used and not Glioblastoma multiforme.

Reply: According to the suggestion of the reviewer, GBM was replaced with glioblastoma.

Q3. Based on typo by me (i.e. mea culpa): Primary glioblastoma are more common than secondray, by far! (90% to 10%, add refernce)

Reply: This was edited according to the suggestion of the reviewer, and a new reference (no. 6) was included as support.

Q4. Page 6/7 there is a paragraph in bold which shouldn't be.

Reply: We thank the reviewer for pointing out this editing oversight, which was corrected accordingly.

Reviewer 2 Report

This is the era of immunogenomic and personalized medicine. The immunologists are struggling to understand and learn genomics. In some reviews, authors with different backgrounds are trying to describe the successes of ”immunotherapy in the marriage between genomics and immunology”.

I believe that the authors must strive to describe briefly at least the link between genomics and immunology/immunotherapy.

Author Response

Q1. This is the era of immunogenomic and personalized medicine. The immunologists are struggling to understand and learn genomics. In some reviews, authors with different backgrounds are trying to describe the successes of ”immunotherapy in the marriage between genomics and immunology”. I believe that the authors must strive to describe briefly at least the link between genomics and immunology/immunotherapy.

Reply: According to the suggestion of the Reviewer, we added a paragraph (section 7) of about 400 words dealing with the link between genomic and immunologic markers in glioma, highlighting the role of “the marriage between genomics and immunology” in the context of glioma biomarkers and therapy.